Peach-Morchella intercropping mode affects soil properties and fungal composition

Song Haiyan 1 2
Chen Dong 1
Sun Shuxia 1
Li Jing 1
Tu Meiyan 1
Xu Zihong 1
Gong Ronggao 2
Jiang Guoliang jgl22@hotmail.com 1
1 Horticulture Research Institute, Sichuan Academy of Agricultural Sciences & Key Laboratory of Horticultural Crop Biology and Germplasm Creation in Southwestern China of the Ministry of Agriculture and Rural Affairs , Chengdu , China
2 College of Horticulture, Sichuan Agricultural University , Chengdu , China
Keller Nancy
Electronic publication date: 2021 Jul 12
Publication date: 2021
Volume: 9
Electronic Location ID: e11705
Received 2021 Feb 18; Accepted 2021 Jun 8
Copyright: ©2021 Song et al.
Copyright year: 2021
Copyright holder: Song et al.
License: This is an open access article distributed under the terms of the Creative Commons Attribution License, which permits unrestricted use, distribution, reproduction and adaptation in any medium and for any purpose provided that it is properly attributed. For attribution, the original author(s), title, publication source (PeerJ) and either DOI or URL of the article must be cited.
License URL: https://creativecommons.org/licenses/by/4.0/

Keywords: Peach orchard, Intercropping, Morchella, Fungual diversity, Soil physicochemical property, Enzyme activity

Funding: National Peach Industry Technical System CARS-31-Z-12 Tackling Key Problems of Crop and Livestock Breeding in Sichuan Province During the 13th Five-year Plan 2016NYZ0034 Innovation Capability Upgrading Project of Sichuan’s Financial Department 2016ZYPZ-019 Sichuan Youth Science and Technology Innovation Research Team 20CXTD0041 Key Research and Development Support Plan in Chengdu 2020-YF09-00065-SN This work was funded by National Peach Industry Technical System (CARS-31-Z-12), Tackling Key Problems of Crop and Livestock Breeding in Sichuan Province During the 13th Five-year Plan (2016NYZ0034), Innovation Capability Upgrading Project of Sichuan’s Financial Department (2016ZYPZ-019), Sichuan Youth Science and Technology Innovation Research Team (20CXTD0041), Key Research and Development Support Plan in Chengdu (2020-YF09-00065-SN). The funders had no role in study design, data collection and analysis, decision to publish, or preparation of the manuscript.

==============================
Objective

This study aims to explore a three-dimensional planting mode in orchards and provide theoretical basis for the efficient peach-Morchella planting and soil management after Morchella cultivation.

Methods

Next-generation sequencing was performed to investigate the variations in soil physicochemical properties, enzyme activities and fungal composition under peach-Morchella intercropping for one year and two years, by using the soil without peach-Morchella intercropping as the control group.

Results

Peach-Morchella intercropping decreased the soil bulk density, and significantly increased the maximum field capacity, non-capillary porosity and total porosity, organic matter, available potassium and available zinc, which together improved soil structure and soil fertility. Besides, the intercropping mode obviously enhanced soil enzyme activities and mineral absorption and transformation in peach orchard soils. The intercropping also resulted in a decline of soil fungal diversity, and the 2-year soil samples were of higher abundance of Zygomycota. More importantly, peach-Morchella intercropping elevated the yields of both peach and Morchella, bringing about obviously higher economic benefits.

Conclusion

Continuous peach-Morchella intercropping improves the soil structure and fertility while decreases soil fungal diversity, which can contribute to greater economic benefits of the peach orchard. Our findings shed new light on the intercropping-fungus-soil relationship, and may facilitate the further development of peach-Morchella intercropping.

Introduction

The Longquanyi District of Chengdu City has a long history of peach planting. However, the management of peach orchards is largely neglected during the slack season, which causes a serious waste of orchard soils (Liu & Zhu, 2009). With the accelerated urbanization, the farmland in China is sharply decreasing. Thus, it is extremely urgent to improve the efficiency of land use (Liu, 2013), for which proper intercropping may be an effective and feasible way (Zhang et al., 2017; Jiang et al., 2017). It has been suggested that monoculture of fruit tree tend to cause some serious problems such as soil erosion, frequent pest and disease occurrence. (Nong, 2017). For these reasons, orchard intercropping, a sustainable development practice that integrates the advantages of agriculture and forestry, has become a popular mode in ecological agriculture. As reported, orchard intercropping can improve soil fertility, water and soil conservation, ecological environment and fruit yield and quality (Zeng, 2009). In recent years, the peach industry in Longquanyi has been declining due to some problems such as cultivar aging and degradation, resulting in lower income of fruit farmers. Fruit-vegetable or fruit-mushroom intercropping in orchards during the slack season are considered to improve the land use efficiency and farmers’ income in this region.

As a valuable edible and medicinal fungus, Morchella is rich in proteins, amino acids, peptides, vitamins and other nutritious components with anti-cancer, anti-fatigue and immunoregulatory activities (Royse & May, 1990; Dai & Yang, 2008; Li, 2018). Currently, the techniques of Morchella cultivation have been relatively well developed, and orchard intercropping of Morchella has been realized. It has been reported that the fruit body of Morchella cultivated with orchard intercropping mode has a higher total amino acid content than that cultivated with conventional field planting (Wei et al., 2020). Fruit-mushroom orchard intercropping has recently become a research topic of great concerns. Chen et al. (2012) conducted intercropping of Pleurotus ostreatus and pear tree, which was found to obviously increase the microbial community in 0–20 cm soil layer, as well as significantly improve the soil fertility and fruit quality. Intercropping patterns would significantly affect soil fungal diversity, changing soil health status, especially for mushroom cultivation (Shen et al., 2009). To promote fruit body occurrence, mushrooms become the dominant fungus in the soil, and compete with other fungal communities, creating a suitable environment. Zhang et al. (2015) and Nong (2017) studied citrus-Stropharia mushroom intercropping system and observed significant increases in soil carbon and nitrogen content, further proving that Stropharia mushroom could promote the formation and mineralization of active organic carbon and various nitrogen sources. In addition, the kiwifruit orchard intercropping pattern was reported to improve the soil permeability and soil aggregate structure, contributing to loose soils and better growth and development of kiwifruit (Lai et al., 2019). Besides, intercropping of edible fungi including Auricularia sp. and Volvariella volvacea in kiwifruit orchards could improve the yield and quality of fruit, bringing about high ecological and economic benefits (Tian & Peng, 1993; Su & Liu, 2014). However, there has been no report on peach-Morchella intercropping yet. In this study, peach-Morchella intercropping was carried out in the peach-planting base of Longquanyi, Chengdu since 2015, and the soil physicochemical properties, enzyme activities and fungal composition were analyzed to investigate the three-dimensional planting mode in orchards and provide theoretical basis for better peach-Morchella intercropping and soil management after Morchella cultivation.

Materials and Methods

Morchella cultivation

The Morchella cultivar Chuan Yangdujun No. 1 was provided by Soil and Fertilizer Institute, Sichuan Academy of Agricultural Sciences. The mushrooms were cultivated in the peach-planting base of Changsong specialized cooperatives in Longquanyi, Chengdu (N30°31′44.55″, E104°17′49.46″) with the soil-covering mode. The base was built in 2012 with a subtropical humid climate and Wanhujing as the main peach cultivar. The peach trees were planted with tree distance of 2.5 m and row distance of 5 m. Morchella was cultivated in the idle fields between peach trees. The seeds were sown in November with the amount of 300–500 g  per m−2, and covered with 2–3 cm thick soil for heat and water preservation. Afterwards, a shelter with arched plastic and shading net was constructed to supply a suitable environment for Morchella growth. About 2 weeks later when the hypha grew out of the soil, the transformation bags were placed in the field with a density of five bags per m−2 for nutrition supply. Then, Morchella form fruit bodies at appropriate time. The harvest time was in February to late March of the next year. After the harvest, the transformation bags were cut and dried for a month, and then the remaining materials inside were poured into the peach-Morchella intercropping fields.

Soil sample collection

A total of three soil treatments were conducted, including one-year peach-Morchella intercropping, two-year peach-Morchella intercropping and the control group without peach-Morchella intercropping (the monoculture peach), and the corresponding soil samples were designated as PM-1, PM-2 and PM-CK, respectively. The PM-2 soil samples were collected from the field where Morchella was intercropped in the peach orchard from 2015 to 2016, while the PM-1 soil samples were collected from the peach orchard with Morchella intercropping just in 2016. Each treatment included three replicates, with each replicate comprising 10 peach trees in a field area of about 150 m2. The distance between each treatment was more than 5 m. After the harvest of Morchella in April, 2017, three sampling sites were randomly selected along a S-shaped curve in each treatment area. Each sampling point was more than 1 m away from the peach tree trunk, and the distance between any two adjacent sampling points was no less than 5 m. Soil samples at the depth of 0–20 cm were collected after the removal of surface litter layer. The physical properties of the soil were measured with samples collected by ring knife and aluminum box sampling. Additional samples were then taken at each point for chemical properties and molecular biological analysis and soil fungal diversity.

Physicochemical determination of soil samples

Soil samples from different treatments were collected and the physicochemical properties were determined, including soil bulk density, maximum field capacity, capillary capacity, noncapillary porosity, capillary porosity, total porosity, moisture content, soil specific gravity, pH value, organic matter, total nitrogen, total phosphorus, available nitrogen, available phosphorus, available potassium and available zinc. Determination of soil bulk density, maximum field capacity, capillary capacity, noncapillary porosity, capillary porosity, total porosity, moisture content and soil specific gravity was performed by referring to NYT 1121.4-2006 (Measurement method from the popularized agricultural standards). pH value was tested using a pH510 table acidometer (EUTECH) based on NYT 1377-2007. Organic matter, total nitrogen, total phosphorus, available nitrogen, available phosphorus, available potassium and available zinc were determined according to the methods reported by Ji (2005), Tang et al. (2009), Zhang (2008), Wang (2008) and Gawryluk, Wyupek & Pawe (2020).

Soil enzyme activity determination

Potassium permanganate ultraviolet spectrophotometry was used to determine the catalase activity (Yang et al., 2011). About 2.00 g soil was weighed and put into a 100 mL triangle flask, which was added with 40 mL of distilled water and then 5 mL of 0.3% H2O2, followed by shaking for 20 min. The samples were then added with 1 mL of saturated alum, immediately filtered and put to a triangle flask containing five mL of 3 N sulfuric acid. After draining, 25 mL of filtrate was absorbed and titrated to purple with 0.1 N potassium permanganate solution. At the same time, a soilless control was performed. DNS colorimetry was used to determine the sucrase and cellulase activity (Li & Zheng, 2016). About 2.00 g of fresh soil was put into a 50 mL triangle flask and injected with 15 mL 8% sucrose solution, 5 mL pH 5.5 phosphate buffer and 0.25 mL toluene. The sample was shaken fully and put into an incubator, cultured at 37 °C for 24 h, and then taken and filtered quickly. Then, 1 mL of filtrate was taken and put into a 50-mL capacity flask, followed by the addition of 3 mL DNS solution and heating in a water bath with boiling water for 5 min. Then, the volumetric flask was moved to tap water and cooled for 3 min. The solution was orange yellow due to the formation of 3-amino-5-nitrosalicylic acid, and was diluted to 50 mL with distilled water. Colorimetric measurement was then performed on a spectrophotometer at 508 nm wavelength. In order to eliminate the errors caused by the original sucrose and glucose in the soil, a control without substrate (without sucrose) should be performed for each soil sample, and a control without soil should be conducted for the whole experiment. For cellulase activity, about 10 g of soil was put into a 50 mL triangle flask, which was added with 1.5 mL toluene, shaken well and allowed to stand for 15 min, followed by the addition of 5 mL 1% carboxymethyl cellulose solution and 5 mL pH 5.5 acetate buffer, and then culturing in a 37 °C incubator for 72 h. After culturing, the sample was filtered and 1 mL of filtrate was taken for colorimetric measurement with the standard curve. In order to eliminate the errors caused by the original sucrose and glucose in the soil, a control without matrix should be performed for each soil sample, and a control without soil should be conducted for the whole experiment. The urease activity was investigated by indophenol blue colorimetry following the descriptions of Huang, Li & Zhang (2012). About 5 g of soil was weighed and put into a 50-mL triangle flask, followed by the addition of 1 mL toluene and shaking, and then addition of 10 mL 10% urea solution and 20 mL citrate buffer solution (pH 6.7) after 15 min. The mixture was well shaken and incubated in a 37 °C incubator for 24 h. After culturing, the mixture was filtered, and then 1 mL of filtrate was taken and added to a 50-mL volumetric flask, followed by the addition of four mL sodium phenol solution and three mL sodium hypochlorite solution and shaking. After 20 min, the color was developed and the volume was fixed. Colorimetric measurement was conducted at 578 nm wavelength of the spectrophotometer within 1 h. Then, excel and SPSS13.0 were used for statistical analysis, and the least significant difference method (P < 0.05) was used for difference analysis.

DNA extraction, PCR amplification and MiSeq sequencing

More than 500 mg of soils for each sample was for DNA extraction with three biological replications in each treatment for the sake of methodological reproducibility. The cetyltrimethyl ammonium bromide method (CTAB) was employed to isolate total DNA from the soil samples by referring to Li et al. (2017), and a UV spectrophotometer from Eppendorf, Bio. Photometer was used to measure the DNA purity and concentration. Then, an appropriate amount of available sample was taken into the centrifuge tube and diluted to 1 ng /µL with sterile water. The diluted genomic DNA was used as the template, and specific primers with barcode were used for PCR amplification according to the sequencing region. The fungal gene-specific primers were ITS5-1737F (GGAAGTAAAAGTCGTAACAAGG) and ITS2-2043R (GCTGCGTTCTTCATCGATGC). To ensure the efficiency and accuracy of PCR amplification, buffer and high fidelity enzyme (New England Biolabs) were used. The PCR amplification was performed by Beijing Novogene Biotechnology Co., Ltd (Beijing, China) with the conditions described by Oros-Sichler & Kornelia (2013). The PCR products were fully and equally mixed based on each concentration, and then detected by electrophoresis with 2% agarose gel. The target product was recovered with the gel recycling kit provided by Qiagen Co., Ltd. Afterwards, the library was constructed with TruSeq®DNA PCR-Free Sample Preparation Kit, which was subsequently quantified by Qubit and Q-PCR. The quantified library was sequenced on HiSeq2500 PE250 (Novogene, Beijing, China).

Sequence and statistical analysis

A sequence alignment tool with QIIME pipeline version 1.7.0 was applied to cluster high-quality sequences with ≥ 97% similarity into OTUs (Edgar, 2010; Caporaso et al., 2010). In this process, the most abundant sequence of each OTU was chosen to as the representative sequence and the relative abundance of the OTUs was calculated. The OTUs would be removed if they had a <0.001% relative abundance of the total sequences across all samples (Bokulich & Mills, 2013). The OTU relative abundance data were subjected to multivariate statistical analysis in R environment (R Core Team, 2016). To visualize the distribution of fungal communities in each treatment, an unconstrained ordination (non-metric multidimensional scaling NMDS) was used based on weighted UniFrac distance using the R vegan package (McArdle & Anderson, 2001; Oksanen et al., 2010). The R VennDiagram package was used to present the numbers of shared OTUs with a Venn diagram (Chen & Boutros, 2011). Besides, a heatmap was drawn to hierarchically cluster and analyze changes (the amount by which each genus deviated in a specific sample from the genus average across all samples) of the 35 most abundant genera using R heatmap package (Kolde, 2015). Two fungal alpha diversity indices including observed OTUs and Shannon were rarefied and calculated based on the smallest library size of the samples.

Results

Variations of soil physicochemical properties in peach orchard

Peach-Morchella intercropping greatly influenced the physicochemical properties of peach orchard soils (Table 1). Among the eight tested soil physical properties, half of them exhibited extremely significant differences compared with those in the control group, while the other four were not obviously different. Specifically, one-year and two-year peach-Morchella intercropping obviously decreased soil bulk density, while significantly increased the maximum field capacity, non-capillary porosity and total porosity compared with the control group. However, capillary capacity was not obviously affected by peach-Morchella intercropping. In addition, all the tested soil chemical properties showed increases in values with the duration of peach-Morchella intercropping except for the pH value. Peach-Morchella intercropping very significantly enhanced organic matter, available potassium and available zinc, particularly two-year intercropping. In addition, two-year peach-Morchella intercropping resulted in extremely significantly higher total nitrogen, total phosphorus and available phosphorus in soils compared with one-year intercropping and the control. Generally, peach-Morchella intercropping evidently improved soil fertility and soil structure with loosened texture and enhanced water holding capacity, but the duration of the intercropping showed no significant effect on the soil physical properties.

Table 1 Soil physicochemical property changes of soil in peach orchard.

NO.	Soil bulk density (g/cm2)	Maximum field capacity (%)	Capillary capacity (%)	Noncapillary porosity (%)	Capillary porosity (%)	Total porosity (%)	Moisture content (%)	Soil specific gravity	
PM-CK	1.47 Aa	29.43 Bb	28.00 Aa	2.55 Bb	40.10 Aa	42.65 Bb	20.98 Aa	2.80 Aa	
PM-1	1.25 Bb	39.97 Aa	27.15 Aa	16.40 Aa	37.50 Ab	51.20 Aa	18.59 Ab	2.54 Ab	
PM-2	1.29 Bb	43.56 Aa	30.25 Aa	16.40 Aa	37.60 Ab	54.00 Aa	20.29 Aa	2.61 Aab	
NO.	pH	Organic matter
g/kg	Total nitrogen
g/kg	Total phosphorus
g/kg	Available nitrogen
mg/kg	Available phosphorus
mg/kg	Available potassium
mg/kg	Available zinc
mg/kg	
PM-CK	7.94 Aa	18.20 Bb	1.35 Bb	0.56 Bb	100.00 Ac	19.70 Bb	165.00 Bc	0.25 Bb	
PM-1	7.88 Aa	24.80 Aa	1.51 Bb	0.68 Bb	115.00 Ab	28.20 Bb	255.00 Ab	0.73 Aa	
PM-2	7.72 Ab	36.37Aa	2.44 Aa	1.21 Aa	134.67 Aa	74.50 Aa	447.67 Aa	1.04 Aa	
Notes.

Excel and SPSS13.0 were used for statistical analysis. Different capital letters showed significant difference at P < 0.01, and different lowercase letters showed significant difference at P < 0.05 between different treatments by the LSD method of a one-way ANOVA. PM-1 the soil of peach-Morchella intercropping for 1 year; PM-2 the soil of peach-Morchella intercropping for 2 years; PM-CK the control group without peach-Morchella intercropping.

Soil enzyme activities in peach orchard

To further explore the variations in soil fertility, we determined the activities of four soil-related enzymes, including catalase, sucrase, cellulase and urease (Table 2). As a result, the activities of all these tested enzymes increased with the duration of peach-Morchella intercropping. Peach-Morchella intercropping resulted in significant increases in the activity of catalase and cellulase in soils, particularly the two-year intercropping, which increased activity of cellulase to 3.6 folds as compared with the control. In addition, the sucrase and urease activities in two-year intercropping soils were significantly higher than those in the control; however, no significant differences were observed between the one-year intercropping soil and the control. Overall, continuous peach-Morchella intercropping could obviously improve soil enzyme activities and facilitate mineral absorption and transformation in peach orchard soils.

Table 2 Soil enzyme activities in peach orchard.

NO.	Catalase (U/g)	Sucrase (U/g)	Cellulase (U/g)	Urease (U/g)	
PM-CK	7.69 Bb	1.10 Bb	0.05 Bc	0.73 Ab	
PM-1	9.25 Aa	1.20 Bb	0.09 Ab	0.82 Aab	
PM-2	9.14 Aa	2.17 Aa	0.18 Aa	0.92 Aa	

Taxonomy-based analysis of fungal community

The soil samples were sequenced and the sequences were clustered into OTUs. As a result, the soil samples from one-year peach-Morchella intercropping were of the largest number of OTUs with an average of 870. However, the soil samples from two-year intercropping had an average of 583 OTUs, which was obviously smaller than that of other two groups. Besides, a total of 273 common fungal OTUs were found in the soils among the three groups (Fig. 1). Among the OTUs classified at the phylum level, Ascomycota and Zygomycota were detected in all groups. As shown in Fig. 2, the most abundant phylum was Ascomycota, accounting for 70.81% of the total fungal sequences on average, followed by Zygomycota (26.13% on average). The relatively less dominant phyla included Basidiomycota, Chytridiomycota, Glomeromycota and Neocallimastigomycota. It is noteworthy that the two-year intercropping soil samples had lower abundance of Ascomycota and higher abundance of Zygomycota relative to other two groups.

To reveal the fungal diversity of different soil samples at the genus level, the dominant genera were clustered in a heatmap (Fig. S1). As a result, 35 genera were clustered and they belonged to the phyla of Ascomycota, Zygomycota and Basidiomycota, accounting for 80.00%, 11.43% and 8.57%, respectively. These genera included Acremonium, Cercophora, Aspergillus, Cladosporium and Tuber. In addition, the differences in dominant genera in different soil samples were investigated. As a result, each group had its specific fungal genera. In particular, one-year intercropping soils had higher abundance of 12 genera such as Acremonium, Cercophora, Aspergillus, Lophodermium and Staphylotrichum, while two-year intercropping soils had higher abundance of eight genera such as Gilbertella, Morchella, Scytalidium and Mortierella. Nevertheless, Trichoderma and Verticillium showed dominance in none of the three soil samples. Hence, it can be inferred that the duration of peach-Morchella intercropping affected the fungal composition in soils to some extent.

Figure 1 Venn diagram showing the number of shared OTUs between different soil samples of peach-Morchella intercropping.

Figure 2 OTU average relative abundances of the major fungal phyla in the soil samples of peach-Morchella intercropping.

Fungal diversity

Two fungal alpha diversity indices were investigated between the soil samples including observed species and Shannon value (Fig. 3). The largest number (423) of observed species was found in the control group, which was similar to that in the one-year intercropping soils but apparently larger than that in two-year intercropping soil samples (301). Similarly, the control group had the greatest Shannon value (5.01), followed by the one-year and two-year intercropping soils. Moreover, the fungal beta diversity was evaluated by nonmetric multidimensional scaling ordination (Fig. S2). The three soil samples were obviously separated in different quadrants. The control and one-year intercropping soil samples were closer on the first axis, while the one-year and two-year intercropping soils were closer on the second axis, indicating their similar fungal communities. Through the whole fungal diversity analysis above, it was further confirmed that continuous peach-Morchella intercropping would result in a decline of soil fungal diversity.

Figure 3 Boxplot showing fungal alpha diversity indices.

Economic benefits

To reveal the effects of continuous peach-Morchella intercropping on the economic benefits of the peach orchards, some relevant economic indices were investigated, including peach yield, fresh Morchella yield, cost and net income (Table 3). Specifically, peach-Morchella intercropping was bound to greatly increase the costs like labor, materials and Morchella culture. However, the harvested Morchella fruit bodies brought about higher economic benefits, and the net income of peach orchards with peach-Morchella intercropping was more than three folds that of the control. Furthermore, a shorter duration of intercropping would more significantly increase the yield of Morchella.

Table 3 Economic benefit investigation of peach orchard with peach-Morchella intercropping.

NO.	Peach yield (kg/mu)	Unit price (yuan/kg)	Fresh Morchella yield (kg/mu)	Unit price (yuan/kg)	Fertilizer cost (yuan/mu)	Other costs (yuan/mu)	Net income (yuan/mu)	
PM-CK	1630	7.5	0	150	1150	300	10775	
PM-1	1650	7.5	240	150	1150	9110	38115	
PM-2	1650	7.5	220	150	1150	7100	37125	
Notes.

Mu, a unit of area (=0.0667 hectares);Other costs included cost of labor, materials and culture of Morchella, etc.

Discussion

Variations in soil physicochemical property contribute to soil structure and fertility improvement

The present study reveals that Peach-Morchella intercropping has great effects on the soil physicochemical properties in peach orchards. The soil bulk density was detected to be obviously decreased. Soil bulk density and porosity are generally regarded as important indicators to evaluate soil quality, both of which play certain roles in determining the status of water, gas, heat and biology in the soil, and affect the nutrient supply for crops (Gu et al., 2010). Consistent with the present study, Wang et al. intercropped edible fungi in pear orchards and observed an obvious decline in soil bulk density, which is conducive to the formation of soil aggregates and increase of soil porosity (Wang et al., 2014). In addition, the maximum field capacity, noncapillary porosity and total porosity were significantly elevated by peach-Morchella intercropping, particularly noncapillary porosity, which was 6.43 folds that of the control. Similarly, the substrates of planted Stropharia mushroom increased the total porosity of orchard soils, which effectively elevated the soil permeability in the study of Duan et al. (2019). Therefore, orchard intercropping can promote soil aggregate formation and effective porosity, which will further improve soil water storage capacity (Zuazo et al., 2009).

Soil physical and chemical properties are closely related to each other. For example, soil type and particle size have great influence on the alkali-hydro nitrogen content (Nong, 2017). In addition, the dynamic soil nitrogen cycle is affected by soil texture, soil depth and precipitation (Kebeney et al., 2014). Our study revealed an extremely significantly higher content of total nitrogen in two-year peach-Morchella intercropping soils. It was previously found that fruit-mushroom intercropping could facilitate the formation and accumulation of active organic nitrogen in soils (Yong, 2020; Coser et al., 2012; Suman et al., 2006), which may be attributed to the mushroom substrates. The waste substrates contain rich carbon and nitrogen nutrients, and small-molecular organic and short-chain fatty acids are released in the process of substrate decomposition, which can stimulate soil microorganism activity to enhance soil nitrogen fixation. Besides, the substrate nitrogen mostly exists in an organic form, which can efficiently alleviate the nitrogen loss in the soil (Lu et al., 2011; Dai, 2009; Tosti et al., 2012). Besides, the organic matter, total phosphorus, available phosphorus and available potassium in soils were significantly enhanced by peach-Morchella intercropping in the present study. Phosphorus is an indispensable nutrient for crop growth and development, and is involved in the synthesis of many important compounds. It can improve the resistance of crops against drought, cold and diseases, and potassium deficiency is likely to cause disorders in metabolism during crop growth (Yang et al., 2012). Generally, the increase in available potassium and available phosphorus in this study provided favorable conditions for the growth of peach. Mu and Duan found obvious increases in soil nutrients including organic matter, available phosphorus and available potassium in mushroom-cultivated soils (Mu, 2019; Duan et al., 2019). Furthermore, Gong (2017) also revealed an increase in soil nutrients under an intercropping mode in a time dependent manner, which is similar to the results in the present study. However, the soil pH value decreased with the intercropping in this study, which is inconsistent with the previous report (Jonathan et al., 2013). In general, soil chemical properties such as organic matter, available nitrogen and available phosphorus can represent the soil fertility to some extent (Zhu et al., 2018). The peach-Morchella intercropping in this study provided better growth conditions for peach growth due to greatly enhanced soil fertility by changing the soil physicochemical properties.

Enhancement of soil enzyme activities affects soil fertility

Soil enzymes participate in the regulation of soil nutrient cycling and organic matter degradation and mineralization. Their activities can reflect soil carbon cycling capacity as well as the soil property and fertility level (Li et al., 2012). In the present study, continuous peach-Morchella intercropping enhanced the activities of all the tested soil enzymes, including catalase, sucrase, cellulase and urease. As is reported, catalase is widely present in soils with the capability of relieving the toxic effect of hydrogen peroxide (Yang et al., 2015). Sucrase affects the carbon transformation in soils (Guo & Zhao, 2010), while urease is capable of increasing available nitrogen content in soils by catalyzing and decomposing urea into carbon dioxide, water and ammonia, and the decomposed substances supply available nitrogen nutrients for fruit trees (Dennis, Miller & Hirsch, 2010; Guo et al., 2013). This study applied substrates from the Morchella transformation bag into the soil, which provided a certain amount of organic matter. Organic matter has a great effect on soil enzyme activities (e.g., urease) (Dai & Chen, 1995). As a result, soil enzyme activity is elevated due to stable enzyme formation through the combination of organic matter with free enzymes (He, 2012). The intercropping pattern in the study of Lai et al. resulted in enhanced activities of urease, sucrase and phosphatase, which would accelerate soil maturation (Lai et al., 2019). Furthermore, secretion from mushrooms may increase the species and amount of soil microorganisms and enzymes, promoting soil acidity and nitrogen accumulation in orchard soils (Coelho et al., 2012; Basilikoa et al., 2012; Phillips et al., 2012). Hence, there is an interactive relationship between soil enzyme activity and organic matter. As a result, peach-Morchella intercropping enhanced soil enzyme activities to improve the soil nutrition (e.g., available nitrogen), which finally contributed to higher fertility of orchard soils in the present study.

Continuous peach-Morchella intercropping decreases soil fungal diversity

Continuous peach-Morchella intercropping was found to decrease soil fungal diversity in the present study. It has been demonstrated that different systems of land use are of distinctive fungal taxon (Plassart et al., 2019). Shen et al. (2009) reported that Morchella cultivation could significantly affect the Discomycetes community structure in soils, and the fungal abundance decreased with Morchella cultivation, which is in accordance with our results. It was speculated that Morchella becomes the dominant fungus in the soil, and competitively inhibits other fungal communities. However, some other studies revealed an increase in bacterial and fungal diversity during mushroom intercropping (Yang, 2019; Lai et al., 2019). It is worth noting that the two-year intercropping soil samples in our study had lower abundance of Ascomycota and higher abundance of Zygomycota. The Discomycetes in Ascomycotina and Heterobasidiomycetes in Basidiomycetes were found to be the dominant fungi in Morchella-cultivated soils in the study of Shen et al. (2009). Zhang et al. (2018) detected 53 fungi from the Morchella-cultivated soil, with Pezizomycetes being the dominant class. Moreover, Ascomycota was identified as the predominant phylum, followed by Mortierellomycota and Basidiomycota in the research of Mu (2019). Besides, the abundance of Ascomycota and Basidiomycota was observed to be significantly changed by Ganoderma lucidum cultivation, as reported by Ren et al. (2020). It can be seen that the variations of fungal communities in mushroom-cultivated soils depend on multiple factors such as cultivation environment and species type. Meanwhile, research on the bacterial community in mushroom-cultivated soils revealed that Pseudomonas is an important bacterial genus, which can promote the occurrence of Agaricus bisporus fruit bodies and plays a significant role in Morchella sclerotium formation (Hayes, Randle & Last, 2010). Furthermore, Xiong et al. analyzed the bacterial community structure in Morchella-cultivated soils, and demonstrated that rich bacterial community structure is conducive to Morchella occurrence (Xiong et al., 2015). Hence, research on bacterial community should be carried out to explore more potential advantages of the peach-Morchella intercropping mode. Soil microbial diversity has a direct impact on soil enzyme activities, thus affecting soil nutrient metabolism and soil fertility (Kumar et al., 2016; Klimek et al., 2016). The intercropping system can enhance interspecific interactions of the underground biota, contributing to the improvement of soil nitrogen supply capacity and increase in total nitrogen content (Li et al., 2018; Dai et al., 2015). To further explain how continuous peach-Morchella intercropping affected fungal community and soil propertie, RDA analysis was done including catalase activity (CaA), urease activity (UA), available nitrogen (AN), total potassium (TK), pH value, and differently-treated soil samples with the top 10 fungal genera in abundance (Fig. S3). Results showed that the fungal communities of 1 and 2-year soil samples with peach-Morchella intercropping were significantly affected by available nitrogen, while that of control was affected by total potassium and pH value. Moreover, the abundances of Morchella, Fusarium, Gibberella and Volutella were positively correlated with soil urease activity and available nitrogen, while Chaetomium and Thermomyces were in a positive relationship with catalase activity and pH value, respectively. Hence, continuous peach-Morchella intercropping promoted the changes of soil fungal community and properties, contributing to a higher fertility of peach orchard soils.

Fruit-mushroom intercropping is of ecological and economic significance

Long-term monoculture tends to cause soil degradation, while intercropping can improve soil micro-ecosystem and physicochemical properties in the farmland (Kou et al., 2010; Pariz et al., 2016). In the present study, the oxygen and root exudates from peach trees helped Morchella to decompose substrates and promoted its mycelial growth. In addition, the carbon dioxide released during Morchella growth could increase the carbon source storage for peach trees, and the enzymes and waste substrates produced by Morchella supplied nutrients for peach trees. Thus, a small but friendly biosphere would be formed. Besides, it has been demonstrated that peach sawdust could accelerate the mycelial growth of Hericium erinaceus and Auricularia cornea. Therefore, it can be speculated that substances produced by peach trees could facilitate the growth and development of Morchella (Shen et al., 2020). The present study revealed higher economic benefits of the peach-Morchella intercropping mode. Similarly, intercropping of Stropharia mushrooms or Dictyophora in grape orchards contributed to higher yields of mushrooms and better soil physicochemical properties. More importantly, the yield and quality of grapes were also improved. Moreover, the edible fungi (e.g., Pleurotus ostreatus) planted under the trees were demonstrated to be of higher nutrition in fruit bodies (Huang et al., 2019). Obviously, the mode can realize coordinated development of fruit and mushroom (Yang et al., 2020). To sum up, the fruit-mushroom intercropping mode such as peach-Morchella intercropping will be an important direction for future fruit and mushroom production.

Conclusion

In the present study, peach-Morchella intercropping was conducted in Longquanyi district of Chengdu city. As a result, most of the soil physicochemical properties changed for better, including the soil bulk density, maximum field capacity, non-capillary porosity, total porosity, organic matter, available potassium and available zinc. In addition, continuous peach-Morchella intercropping improved soil enzyme activities but decreased soil fungal diversity. Finally, the soil structure and fertility of the studied peach orchards were significantly improved, which possibly contributed to the yield increase of peach and Morchella. The findings provide a theoretical basis for the efficient peach-Morchella intercropping mode and soil management after Morchella cultivation.

Supplemental Information

Supplemental Information 1 Raw data for Tables 1–3

Click here for additional data file.

Supplemental Information 2 SRA accession numbers

Click here for additional data file.

Supplemental Information 3 Heat map of fungal abundances at genus level in the soil samples of peach-Morchella intercropping

Click here for additional data file.

Supplemental Information 4 Nonmetric Multidimensional Scaling ordination of fungal communities based on weighted UniFrac distance

Each point in the diagram represents a fungal community sample. The closer distance between two points in the ordination space indicates the lower dissimilarity between the microbial community structure of these two samples.

Click here for additional data file.

Supplemental Information 5 RDA analysis between some soil properties and differently-treated soil samples

CaA, catalase activity; UA, urease activity; AN, available nitrogen; TK, total potassium; pH, pH value.

Click here for additional data file.

Additional Information and Declarations

Competing Interests

Author Contributions

DNA Deposition

Data Availability

The authors declare there are no competing interests.

Haiyan Song conceived and designed the experiments, performed the experiments, analyzed the data, prepared figures and/or tables, authored or reviewed drafts of the paper, and approved the final draft.

Dong Chen conceived and designed the experiments, analyzed the data, authored or reviewed drafts of the paper, and approved the final draft.

Shuxia Sun performed the experiments, authored or reviewed drafts of the paper, and approved the final draft.

Jing Li and Meiyan Tu performed the experiments, prepared figures and/or tables, and approved the final draft.

Zihong Xu analyzed the data, authored or reviewed drafts of the paper, and approved the final draft.

Ronggao Gong analyzed the data, prepared figures and/or tables, and approved the final draft.

Guoliang Jiang conceived and designed the experiments, authored or reviewed drafts of the paper, and approved the final draft.

The following information was supplied regarding the deposition of DNA sequences:

Sequences are available at NCBI SRA: PRJNA702601. The individual accession numbers are available in Supplemental Information 2.

https://www.ncbi.nlm.nih.gov/sra/?term=PRJNA702601.

The following information was supplied regarding data availability:

The raw data are available in the Supplemental Files.

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
