# Peer review of "Peach-Morchella intercropping mode affects soil properties and fungal composition"

_PeerJ, doi:10.7717/peerj.11705_

## Round 0.1 · original submission · Minor Revisions

You have two expert reviews who provided good guidance for improving your submission.

Reviewer 1 ·

Basic reporting

The English language looks good, and some errors were existed in the text.
In the introduction, please focus on the advances of the soil microorganism and soil health caused by intercropping patterns.

Experimental design

The experimental design is ok.
Please provide the details of the methods of enzyme activity assays.

Validity of the findings

This manuscript was study the effects of peach-Morchella intercropping on the soil physicochemical properties, enzyme activities and fungal community in an orchard. The results indicated that intercropping reduced soil fungal diversity and improved soil structure and fertility, and suggested intercropping is the efficient planting practice for soil health and farmers’ income. The data is detailed, and the result part is also very sufficient.

Additional comments

Some amendments are suggested as follows:
Line 39, delete “.” After “activities”.
Lines 64-67, I do not understand the planting processes of Morchella from the description.
Line 72, the PM-CK was without peach-Morchella intercropping. Was it from the monoculture peach or Morchella?
Line 78, please descript the sites of soil sampling. How far was the sample from the tree?
Please check the cited format of the reference, such as “Gawryluk, Wyupek & Pawe” in the Line 89, “Huang, Li & Zhang” in the Line 94, and in the Lines 198-200, and so on.
I found the soil organic matter was sharply increased in 1 or 2 years. Generally, the increase of soil organic matter was slow, please explain it.
Lines 180-182, I do not think CK and PM1 were closer on the first axis, while PM1 and PM2 were closer on the second axis, and thus confirmed continuous intercropping result in the decline of the soil fungal diversity.
Please check the “sucrose” and “sucrase”.
Please increase the font size in the Figure 3.
In the discussion, the authors mentioned “Soil physical and chemical properties are closely related to each other”, “Organic matter has a great effect on soil enzyme activities”. I suggest the authors add the correlation analysis between soil physicochemical and biological properties, especially the association between the sol properties and fungal community in order to improve the quality of the manuscript.

Reviewer 2 ·

Basic reporting

In general, the paper sent a clear message and demonstrated an interesting intercropping system of peach and Morchella mushroom, which yielded significant impact on both economic output and soil health. The article provided sufficient background and context regarding the relevance and scientific basis of intercropping systems on fruit crops. I commend the authors for sharing the detailed sequencing and crop yield raw data to support their claims.

Experimental design

The research questions regarding soil health, reflected in soil physico-chemical properties, crop yield, and economical benefits were clearly laid out in the introduction and well-addressed by the results.

Validity of the findings

The author clearly stated the experimental set up and the results are clearly supported by data.

Additional comments

In general, the paper sent a clear message and demonstrated an interesting intercropping system of peach and Morchella mushroom, which yielded significant impact on both economic output and soil health.

There are, however, several places need to be improved before this paper can be accepted for publication:

1.Line 70: Soil sample collection: please explain how far away from the peach trees the soil samples were collected. The difference between bulk soil and rhizosphere microbiome can influence the result and interpretation of the data collected in this study.

2.Line 74: Did you mean “from 2015 to 2017”?

3.Line 86: An brief explanation of what NYT 1121.4-2006 is should be included.

4.Line 100: “…employed to isolate fungal DNA”. I assume that the first step was to extract total DNA in the soil, not just fungal DNA. Only fungal ITS sequences were enriched from total soil DNA in the following step.

5.Table 1: I suggest the authors include standard deviations/standard errors when reporting statistics of the soil physico-chemical properties, and indicate specifically which statistical test was used to access differences between groups here (presumably, one-way ANOVA?)

6.Table 1 and throughout the manuscript: The author should replace “Pure Income” with a more accurate term, such as “Net income”, of “Profit” as this value was calculated as such.

7.Line 178: typo (should be “Fungal” instead of “Fugal”)

8.Line 273-274: “the fungal community is probably an important contributor to the higher fertility of peach orchard soils in this study”. It is surprising that the authors observed a reduction in fungal diversity, especially at year 2 of inter-cropping. I would like to see the authors’ elaboration on the correlation between fungal diversity and soil health, especially in the context that there are previous reports linking the positive correlation between fungal diversity and soil fertility.

9.Also, would the author expect that continuous peach-Morchella inter-cropping would make the soil fungal diversity worst in the long run?

10.Figure S1: it would be informative if the authors can show individual replicate and the hierarchical analysis to see if the same treatments would clusters together.

11.Please elaborate on how inter-cropping and the reduction of general fungal diversity would affect the abundance of pathogenic fungi often found in this system.

---

## Round 0.2 · accepted · Accept

Thank you for meeting the reviewers' comments.